# Transcriptional Analysis of Antrodin C Synthesis in *Taiwanofungus camphoratus* (Syn. *Antrodia camphorate*, *Antrodia cinnamomea*) to Understand Its Biosynthetic Mechanism

**Wei Jia [1,†], Shu-Ping Gai [2,†], Xiao-Hui Li [2], Jing-Song Zhang [1,*] and Wen-Han Wang [1,*]**

[1] Key Laboratory of Edible Fungi Resources and Utilisation (South), Institute of Edible Fungi, Shanghai Academy of Agricultural Sciences, Ministry of Agriculture and Rural Affairs, National Engineering Research Centre of Edible Fungi Shanghai, Shanghai 201403, China; jiawei@saas.sh.cn

[2] College of Food Science and Technology, Shanghai Ocean University, Shanghai 201306, China; gaishuping0@163.com (S.-P.G.); xhli@shou.edu.cn (X.-H.L.)

\* Correspondence: syja16@saas.sh.cn (J.-S.Z.); wangwenhan@saas.sh.cn (W.-H.W.)

† These authors contributed equally to this work.

**Abstract:** Antrodin C, a bioactive component of *Taiwanofungus camphoratus*, exhibits good immunophysiological and antitumour activities, including a broad spectrum of anticancer effects. Exogenous additives can bind to metabolites during the submerged culture of *T. camphoratus* and affect secondary metabolite yields. However, the lack of molecular genetic studies on *T. camphoratus* has hindered the study of the antrodin C biosynthetic pathway. In this study, we conducted a ribonucleic acid-sequencing-based transcriptional analysis to identify the differentially expressed genes involved in the synthesis of antrodin C by *T. camphoratus*, using inositol and maleic acid (MAC) as exogenous additives. The addition of inositol significantly upregulated carbohydrate and sugar metabolism pathway genes (*E3.2.1.14*, *UGDH*, and *IVD*). When MAC was used, amino and nucleotide sugar metabolism and starch and sucrose metabolism pathways were significantly inhibited, and the associated genes (*E3.2.1.14* and *E3.2.1.58*) were also significantly downregulated. The biosynthesis pathway genes for ubiquinone and other terpene quinones (*COQ2*, *ARO8*, and *wrbA*), which may play an important role in antrodin C synthesis, were significantly downregulated. This study advances our understanding of how the additives inositol and MAC affect metabolite biosynthesis in *T. camphorates*. This could be beneficial in proposing potential strategies for improving antrodin C production using a genetic approach.

**Keywords:** *Taiwanofungus camphoratus*; antrodin C; submerged culture; transcriptomic analysis; quantitative PCR

## 1. Introduction

*Taiwanofungus camphoratus* belongs to the class Agaricomycetes and family Polyporaceae and is a medicinal and food fungus unique to Taiwan [1]. Previous studies have shown that *T. camphoratus* exhibits a variety of medicinal activities, such as antitumor [2], antioxidant [3], immunomodulatory [4], anti-inflammatory [5], and hepatoprotective properties [6], which can be attributed to a variety of bioactive components present in *T. camphoratus*. Among these bioactive components, antrodins (A–E) are important active components unique to *T. camphoratus*.

The antrodin class of compounds, which consists of maleic and succinic acid derivatives, were first isolated from the liquid-fermented mycelium of *T. camphoratus* by Nakamura et al. [7]. Antrodin C is the most widely studied antrodin compound, with studies mainly focusing on its biological activity and pharmacology. Antrodin C has been shown to substantially inhibit hepatitis C virus and Lewis lung carcinoma tumour cells, and it

exhibits better inhibition of TGF-β1-induced epithelial–mesenchymal transition and breast cancer cell metastasis [8,9].

Antrodin C is mainly found in the mycelia of *T. camphoratus*. Compared to cultivating mycelia using solid-state fermentation, the submerged culture method allows for the production of a large amount of mycelia or metabolites in a short time [10], and it offers the advantages of a rapid growth rate, ease of regulation, and a reduced risk of contamination [11]. Therefore, promoting antrodin C biosynthesis during submerged fermentation is important. Xia et al. [12] performed response surface analysis to optimise the liquid fermentation medium composition, and this resulted in a significant increase in antrodin C yield, which was 85.8% higher than the yield before optimisation. Zhang et al. [13] designed a novel integrated fermentation system, combining a surfactant and an in situ extractant, which increased the yield of antrodin C. This system was designed to improve the membrane permeability of mycelial cells and enhance the production of antrodin C. In addition, a novel integrated fermentation system combining a surfactant and an in situ extractant was designed in their study. The addition of Tween 80 to the culture medium improved the membrane permeability of mycelial cells, promoting the extracellular secretion of antrodin C. Liu et al. [14] used vegetable oil as the solvent for extraction and fermentation, and added the precursor coenzyme Q0 in the liquid medium to induce the synthesis of active ingredients. This effectively increased the yield of antrodin C analogues (486.01 mg/L) by 272.1% compared with the yield when using uninduced fermentation. In recent years, studies of antrodin C have also focused on increasing the amount of biomass and improving the production of secondary metabolites [15]. However, the mechanism of metabolic regulation of antrodin C remains unclear. In our previous study, we found that the addition of inositol to the liquid medium significantly increased the yield of antrodin C; however, maleic acid (MAC) significantly inhibited its yield [16]. We hypothesised that different additives would stimulate the expression of distinct genes. However, molecular studies related to these metabolic processes are lacking, and the regulatory mechanisms remain unclear.

In this study, we investigated changes in antrodin C production in submerged cultures of *T. camphoratus* using inositol and MAC as exogenous additives, and we determined the optimal concentration of these additives. To elucidate the genes that were differentially expressed in the presence of exogenous additives, we investigated the transcriptional profile of *T. camphoratus* using next-generation sequencing technology. The messenger ribonucleic acid (mRNA) expression levels of potential metabolite-related genes were verified using quantitative polymerase chain reaction (qPCR). The identification of these genes may be useful for analysing the biosynthetic pathways of different metabolites induced by the different additives. The findings of this study provide new ideas and experimental means for the study of the biosynthetic pathway of antrodin C in *T. camphoratus* and are expected to facilitate important breakthroughs in revealing the synthetic pathways and regulatory mechanisms of antrodin C.

## 2. Materials and Methods

### 2.1. Strain and Culture Conditions

*T. camphoratus* strains were obtained from the Institute of Edible Fungi of the Shanghai Academy of Agricultural Sciences (Shanghai, China). The seed and fermentation medium was potato dextrose broth (PDB), purchased from BD Biosciences (San Jose, CA, USA).

Four 0.5 cm$^2$ agar blocks containing *T. camphoratus* mycelia (cultured on potato dextrose agar at 26 °C for 14 d) were transferred to 250 mL Erlenmeyer flasks containing 100 mL of PDB, incubated at 26 °C and 150 rpm for 14 d, and then transferred to 500 mL Erlenmeyer flasks containing 200 mL PDB and cultured under the same conditions for 5 d as liquid spawn.

### 2.2. Determination of Biomass

Liquid spawn (10 mL) was inoculated into 250 mL Erlenmeyer flasks containing 110 mL of PDB and incubated at 26 °C and 150 rpm. After incubation for 0, 2, 4, 6, 8, and 10 d, the biomass amount, residual sugar content, and antrodin C yield were measured.

### 2.3. Determination of Antrodin C Yield with Different Concentrations of Inositol and Maleic Acid

Different concentrations of inositol (0, 0.05, 0.1, 0.15, and 0.2 g/L) were added to 250 mL Erlenmeyer flasks containing 110 mL of PDB. Liquid spawn (10 mL) was inoculated, and the fermentation medium was cultured continuously for 8 d at 26 °C and 150 rpm. The final culture was tested for the amount of biomass and the antrodin C yield.

Different concentrations of MAC (0, 0.05, 0.1, 0.2, and 0.4 g/L) were added to 250 mL Erlenmeyer flasks containing 110 mL of PDB. Liquid spawn of *T. camphoratus* was homogenised under aseptic conditions to obtain a suspension of *T. camphoratus* mycelia, and 10 mL of this mycelial suspension was inoculated into the fermentation medium. The medium was continuously cultured at 26 °C and 150 rpm for 8 d to investigate the effects on mycelial growth and antrodin C production.

### 2.4. Detection of Biomass in Fermentation Broth

The fermentation broth from the final culture was filtered through four layers of gauze, and the obtained mycelia were washed thrice with deionised water and dried at 60 °C until a constant weight (g/L) was reached.

### 2.5. Detection of Residual Sugars in Fermentation Broth

Following the removal of the mycelia, the filtrate was tested for residual sugar content (g/L) using a biosensing analyser (Institute of Biology, Shandong Academy of Sciences, Jinan, China).

### 2.6. Extraction and Determination of Antrodin C Yield

Dried mycelia were extracted with 10 times the volume of 95% ethanol using ultrasonic extraction for 70 min. Subsequently, the mixture was centrifuged at $2057 \times g$ for 10 min at 4 °C. The supernatant was filtered through a 0.22 μm organic microporous filter membrane, and the filtrate was analysed using high-performance liquid chromatography (Waters 2695; Waters, Milford, MA, USA) using a ZORBAX SB-Aq C18 column (4.6 × 250 mm, 5 μm; Agilent, Santa Clara, CA, USA). Detection was performed at 254 nm.

The mobile phase consisted of $CH_3CN:H_2O$ (0–15 min, A 50–100% and B 50–0%; 15–15.5 min, A 100% and B 0%; 15.5–25 min, A 100–50% and B 0–50%). The flow rate was 1.0 mL/min at 35 °C. The antrodin C content was calculated from the peak area using a standard curve, and the yield (mg/L) of antrodin C in the fermentation broth was determined based on the dilution of the sample.

### 2.7. Transcriptome Analysis

The fermentation broth of *T. camphoratus* from three fermentation cultures cultivated for 8 d was filtered through four layers of gauze. The mycelia were rinsed thrice with deionised water and then stored at −80 °C after rapid freezing in liquid nitrogen. Medium without the addition of MAC and inositol was used for the control mycelia (CK) group, liquid medium with MAC added as a precursor was used for the maleic-acid-fermenting mycelia group, and liquid media with inositol added as a precursor was used for the inositol-fermenting mycelia group.

The three sets of samples were re-sequenced by Shanghai Parsonage Biotech (Shanghai, China) using a NovaSeq sequencing platform (Illumina, San Diego, CA, USA). Clean data were obtained by removing adapter reads and those with an average mass fraction less than Q20. The filtered reads were compared with the reference gene (accession number: GCA_022598655.1, NCBI) using the upgraded HISAT2 v2.2.0 software, TopHat2 (http://tophat.cbcb.umd.edu). Expression saturation was analysed using RSeQC v2.6.3.

Read-count values were compared for each gene using HTSeq v0.6.1 statistics for raw gene expression. Expression levels were normalised based on fragments per kilobase of transcripts per million fragments (FPKM) values. Principal component analysis (PCA) and correlation analysis were performed based on FPKM values. Differential gene expression analysis was performed using DESeq, and differential expression levels and *p*-values were calculated for each gene. Differentially expressed genes (DEGs) were identified by comparing the expression levels between several comparison groups. DEGs were annotated and functionally analysed using the Gene Ontology (GO; Gene Ontology Resource) and Kyoto Encyclopaedia of Genes and Genomes (KEGG: Kyoto Encyclopedia of Genes and Genomes) databases.

*2.8. qPCR Analysis*

Total RNA was extracted from the submerged fermentation mycelia of *T. camphoratus* using the TRIzol method [17]. To obtain complementary deoxyribonucleic acid (cDNA) from total RNA, reverse transcription was performed using a PrimeScript™ II 1st Strand cDNA Synthesis Kit (Takara, Dalian, China). Reverse transcription-qPCR was performed using reverse-transcribed cDNA as the template. The $2^{-\Delta\Delta Ct}$ [18] calculation method was applied to quantify the transcription level of genes using *18S rRNA* levels of *T. camphoratus* as an internal reference. The primer sequences for the genes analysed are listed in Table S1.

*2.9. Statistical Analysis*

Three replicates were performed per experimental group, and the data are expressed as the mean ± standard deviation. One-way analysis of variance was performed using SPSS Statistics 26 (IBM, Armonk, NY, USA), and differences were considered statistically significant at $p < 0.05$. Origin v2021 software was used for plotting.

## 3. Results

### 3.1. Biological Characterisation of T. camphoratus

Culture time has a substantial influence on the biological characteristics of *T. camphoratus*, especially the amount of biomass. Liquid spawn cultured for 5 d was inoculated into the fermentation broth of conventional fermentation media. The initiation of the first transfer was marked as day 0. The effects of different fermentation periods on the amount of biomass, residual sugar content, and antrodin C yield of *T. camphoratus* cultures were analysed. As shown in Figure 1, the amount of residual sugar did not change significantly during the first 6 d, but decreased markedly from day 6, indicating that glucose consumption increased with culture time. The amount of biomass of *T. camphoratus* increased with incubation time, reaching logarithmic growth on day 4. When incubated until day 8, the amount of biomass reached a maximum of 1.23 g/L and then began to decline. The trend for antrodin C content was similar to that of biomass, reaching a maximum of 52.76 mg/L on day 8 and then declining sharply. This may be attributed to mycelial autolysis after day 8, resulting in a decrease in the amount of biomass. Given that antrodin C is mainly produced within the mycelia of *T. camphoratus*, its content also began to decrease after day 8. Therefore, in subsequent studies, a fermentation time of 8 d was used.

### 3.2. Effect of Different Additives on Biomass and Antrodin C Yield of T. camphoratus

Different concentrations of exogenous additives were added to study their effects on mycelial growth and antrodin C production. Figure 2 shows that, compared with the control group, the addition of MAC and inositol had no significant effect on the biomass of *T. camphoratus* but had a significant effect on the yield of antrodin C. The addition of MAC significantly inhibited the biosynthesis of antrodin C compared with its level of biosynthesis in the CK group. When the concentration of MAC was 0.1 g/L, the yield of antrodin C was the lowest (25.31 mg/L), which was 40.5% lower than the yield in the control group (42.55 mg/L, $p < 0.05$). This shows that antrodin C production during submerged fermentation by *T. camphoratus* may not be directly related to mycelial growth.

This result was similar to the result reported by Zhang et al. on the effect of surfactant incorporation in *T. camphoratus* [13]. Contrary to the results observed after MAC addition, the yield of antrodin C in *T. camphoratus* was lowest when inositol was not added. The production of antrodin C increased with the increase in inositol concentration, reaching a maximum (19.76 mg/L) at an inositol concentration of 0.15 g/L, which was a significant increase of 63.98% over the production of antrodin C in the control group (12.05 mg/L). A further increase in the inositol concentration led to a decrease in antrodin C production. This may have been due to the excessive destruction of *T. camphoratus* mycelia by high concentrations of inositol.

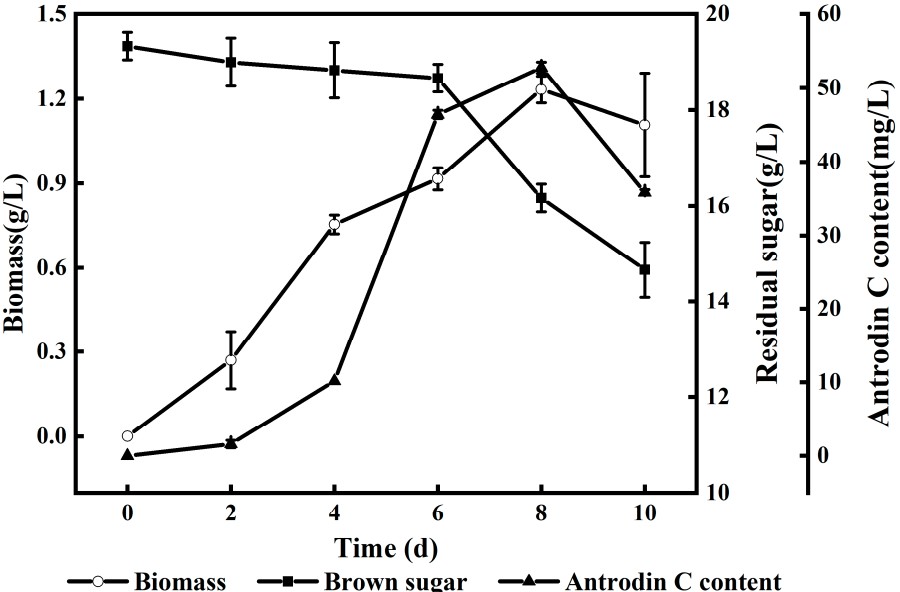

**Figure 1.** Metabolic pattern of the fermentation products of *Taiwanofungus camphoratus* under different culture periods.

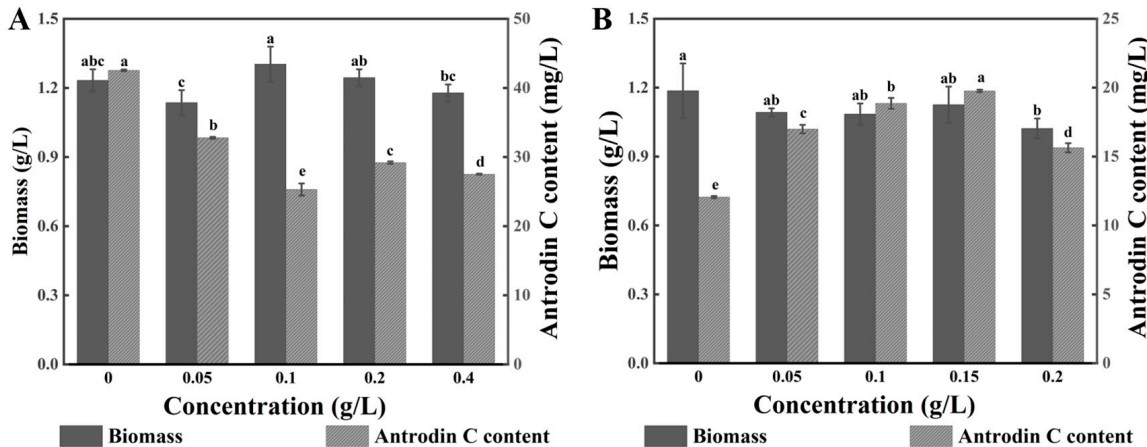

**Figure 2.** Effect of different concentrations of exogenous additives on biomass and antrodin C production in *T. camphoratus*. (**A**) Maleic acid, MCA; (**B**) inositol. Different lowercase letters (a, b, c, d, and e) indicate significant differences ($p < 0.05$).

Based on these results, the addition of 0.1 g/L MAC and 0.15 g/L inositol was used in subsequent experiments, and sequencing analysis was performed to analyse the transcriptome and elucidate the changes in gene expression between the treatment and control groups.

### 3.3. Descriptive Analysis of RNA Sequencing Data

The NovaSeq platform was used to sequence the samples. Raw data were filtered, and adapter sequences and low-quality reads were discarded. In our pre-experiment, the whole genome of *T. camphoratus* mononucleosomes obtained by protoplast technology was sequenced, and the genome size of *T. camphoratus* was found to be 32.96 Mb. Mapping against the reference genome (GCA_022598655.1) showed a sequence similarity of 99.42%, which indicated that our strains were identical to the reference strain. Table 1 shows the statistics of the clean and mapped reads from the RNA sequencing (RNA-seq) analysis. For each of these three samples, we obtained 40–49 million clean reads and 604–744 million high-quality sequence bases. The Q20 and Q30 scores were 97% and 93%, respectively. Of the total clean reads in the three sample groups, the overall sequence comparison was in the range of 97.96–98.36%. Additionally, 94.40–96.17% of the reads were uniquely mapped, and 3.83–5.60% were mapped to multiple loci, indicating that the reference genome was appropriate for subsequent testing.

**Table 1.** Statistics of filtered RNA-seq data and comparison results.

| Sample | Clean Reads | Clean Data (bp) | Q20 (%) | Q30 (%) | Top Mapped | Multiple Mapped | Uniquely Mapped |
|---|---|---|---|---|---|---|---|
| CK 1 | 44,798,630 | 6,760,298,404 | 97.76 | 93.82 | 43,882,934 (97.96%) | 1,755,369 (4.00%) | 42,127,565 (96.00%) |
| CK 2 | 40,799,084 | 6,153,300,013 | 98.09 | 94.5 | 40,129,578 (98.36%) | 2,245,947 (5.60%) | 37,883,631 (94.40%) |
| CK 3 | 41,295,462 | 6,226,806,032 | 97.87 | 94.15 | 40,531,757 (98.15%) | 1,833,723 (4.52%) | 38,698,034 (95.48%) |
| IO 1 | 46,860,468 | 7,069,852,321 | 97.63 | 93.57 | 45,980,182 (98.12%) | 1,767,746 (3.84%) | 44,212,436 (96.16%) |
| IO 2 | 45,405,964 | 6,851,073,561 | 98.08 | 94.61 | 44,576,558 (98.17%) | 1,706,534 (3.83%) | 42,870,024 (96.17%) |
| IO 3 | 46,122,616 | 6,959,518,905 | 97.74 | 93.81 | 45,247,681 (98.10%) | 1,781,445 (3.94%) | 43,466,236 (96.06%) |
| MAC 1 | 40,041,196 | 6,041,792,837 | 98.05 | 94.6 | 39,246,233 (98.01%) | 1,873,489 (4.77%) | 37,372,744 (95.23%) |
| MAC 2 | 42,802,110 | 6,458,223,523 | 97.82 | 93.98 | 41,937,635 (97.98%) | 2,024,335 (4.83%) | 39,913,300 (95.17%) |
| MAC 3 | 49,487,160 | 7,449,231,565 | 98.13 | 94.75 | 48,535,741 (98.08%) | 2,423,780 (4.99%) | 46,111,961 (95.01%) |

### 3.4. Sample Reproducibility and DEGs

Based on the FPKM values from the expression database (Table S2), PCA and correlation analysis (Figure 3) were performed to compare gene expression levels among the three groups. Compared with the inositol group, the CK and MAC groups showed tight clustering (Figure 3A), and the distance between the inositol and CK groups was large, indicating significant differences in gene expression levels between these two groups. At the same time, the three groups were distinguishable, indicating significant differences among them (Figure 3B).

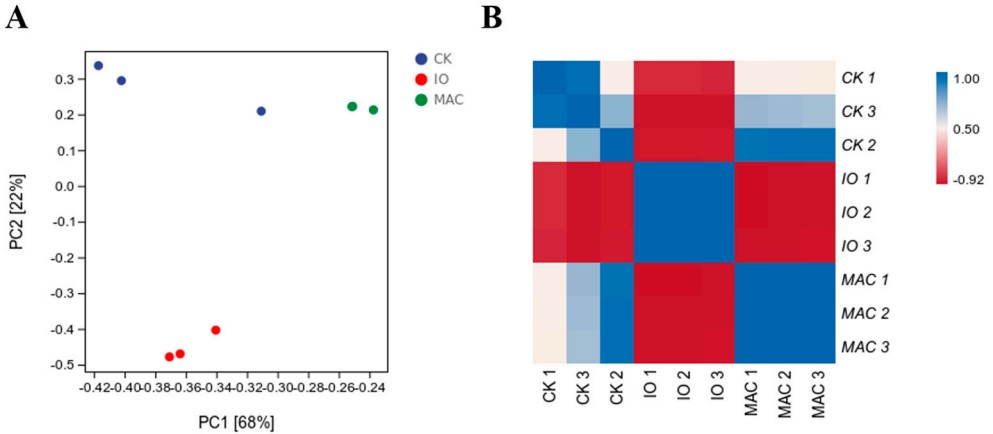

**Figure 3.** Statistical analysis of sample repeatability and differentially expressed genes. (**A**) Principal component analysis; (**B**) correlation analysis.

*3.5. Identification of DEGs*

Hierarchical clustering based on gene expression patterns was used to identify functionally enriched clusters (Figure S1), with red indicating increased transcript abundance and blue indicating decreased transcript abundance. Figure S1 shows that the addition of different additives induced differences in *T. camphorates* gene expression levels. Volcano and Wayne plots were used to analyse the DEGs among the three groups (Figure 4).

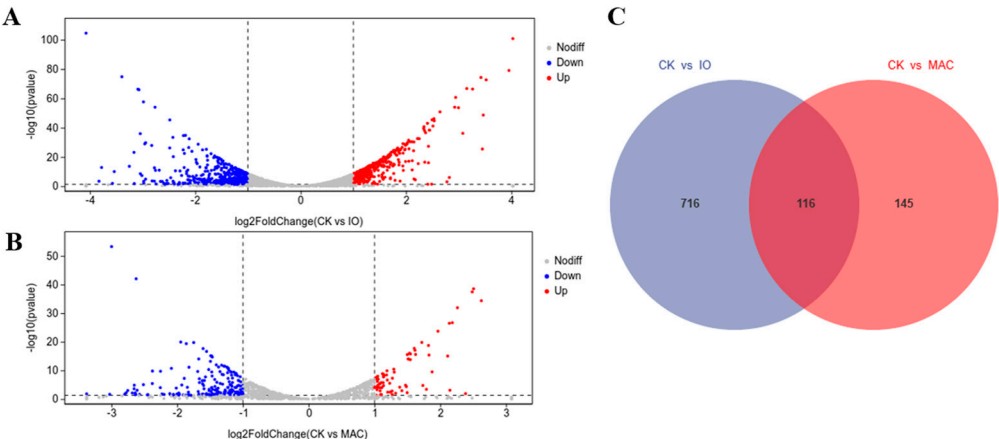

**Figure 4.** Volcano plots (**A**,**B**) and Wayne plots (**C**) of differentially expressed genes between the three groups. CK, control; IO, inositol; MAC, maleic acid.

As shown in Figure 4A,B, the DEGs differed between the CK and MAC groups and between the CK and inositol groups. A total of 832 DEGs were identified between the CK and inositol groups and 261 between the CK and MAC groups (Figure 4C). In total, 716 DEGs were identified only between the CK and inositol groups, which was similar to the results presented in Figure 3A. In contrast, only 145 DEGs were identified between the CK and MAC groups.

*3.6. Functional Analysis of DEGs Based on GO Categories*

Functional classification enrichment analysis was performed on the DEGs covering the following three gene ontology (GO) categories: biological processes, cellular components, and molecular functions. The results are shown in Figure 5. DEGS between the CK and inositol groups were enriched in two major classes: cellular components and biological processes (Figure 5A; Table S3). In the category of biological processes, the main enriched processes were mitotic cell cycle and DNA replication and elongation, while nuclear replication, protein–DNA complexes, nuclear chromosomes, and condensing complexes were the significantly enriched cellular components.

Three important aggregates were identified in the molecular functions of the DEGs between the CK and MAC groups (Figure 5B; Table S4). In the category of biological processes, protein folding, response to heat, catabolic processes, and biosynthetic processes were significantly enriched, and a total of 11 DEGs were classified as extracellular region and cell surface in the cellular component category. Of the molecular function terms, unbound proteins, aryl alcohol dehydrogenase (NAD$^+$) activity, hydrolase activity, and hydrolysis of O-glycosylated compounds were enriched.

*3.7. Functional Analysis of DEGs Based on KEGG Pathways*

KEGG enrichment analysis was used to identify the metabolic pathways associated with the DEGs. Figure 6 shows that 62 (Table S5) and 28 (Table S6) metabolic pathways were enriched in the inositol and MAC groups, respectively, compared to the CK group. DEGs in the inositol group were mainly enriched in DNA replication; cell cycle-yeast; pentose and glucuronate interconversions; ABC transporters; homologous recombination; nucleotide excision repair; ascorbate and alternate, purine, caffeine, glycerolipid, glutathione, and

amino sugar and nucleotide sugar metabolisms; meiosis-yeast; and protein processing in the endoplasmic reticulum. In contrast, DEGs in the MAC group were mainly enriched in protein processing in the endoplasmic reticulum; steroid biosynthesis; amino sugar and nucleotide sugar, glyoxylate and dicarboxylate, tryptophan, and starch and sucrose metabolism; and ubiquinone and other terpenoid quinone biosyntheses.

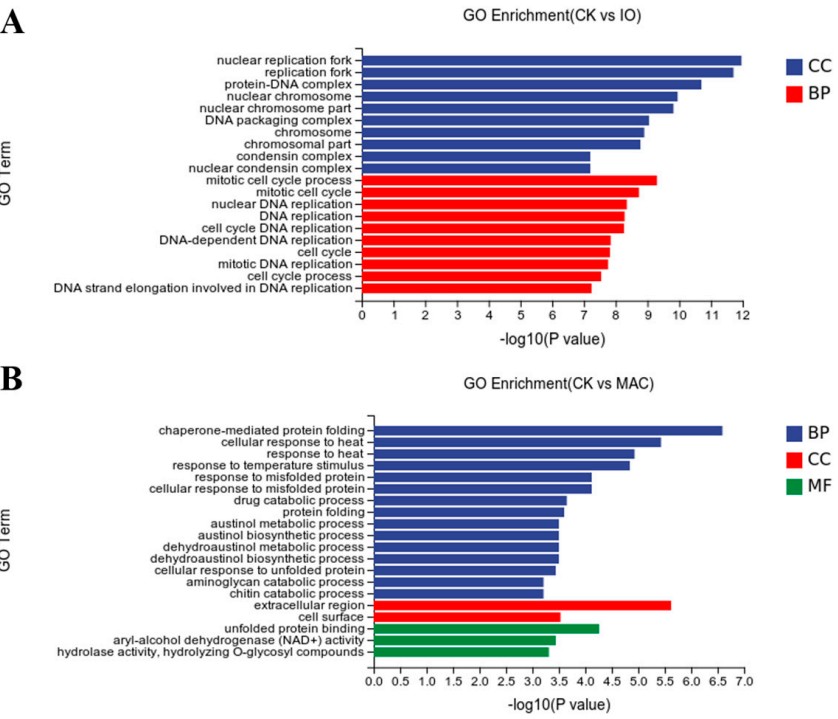

**Figure 5.** Gene ontology classification enrichment analysis of differentially expressed genes. The top 20 metabolic pathways enriched in the inositol (IO) (**A**) and maleic acid (MAC) (**B**) groups compared to the control (CK) group are shown. BP, biological process; CC, cellular component; MF, molecular function; GO, Gene Ontology.

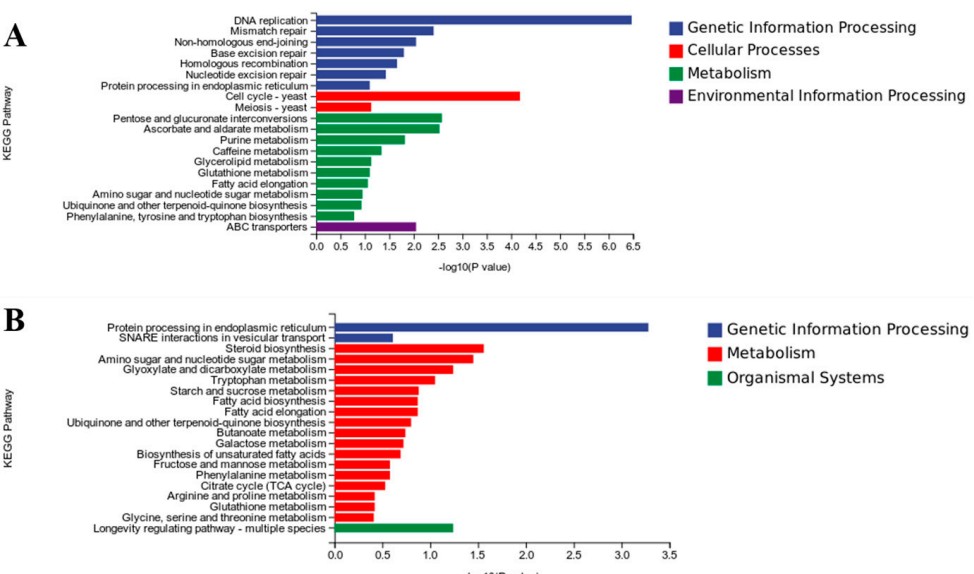

**Figure 6.** Kyoto Encyclopedia of Genes and Genomes (KEGG) pathway classifications of differentially expressed genes. The top 20 metabolic pathways enriched in the inositol (IO) (**A**) and maleic acid (MAC) (**B**) groups compared to the control (CK) group are shown.

To further elucidate the differential expression of genes associated with antrodin C synthesis, the DEGs were categorised into several pathways based on those involved in metabolism (Tables 2 and 3). A total of 89 DEGs between the CK and inositol groups were involved in metabolic pathways (Table S2), while only 24 DEGs between the CK and MAC groups were involved in metabolic pathways (Table S3). Among the seven metabolic pathways, the purine metabolic pathway had the greatest number of DEGs between the CK and inositol groups (Table 2), and up to two DEGs between the CK and MAC groups were enriched in these metabolic pathways (Table 3). Notably, both the inositol and MAC groups were enriched in carbohydrate metabolism and biosynthetic pathways, such as ubiquinone, compared with the CK group, although the genes involved in these pathways differed. *COQ2* and *ARO8*, which are related to the ubiquinone and other terpene-benzoquinone biosynthesis pathways, were differentially expressed between the CK and inositol groups (Table 2), whereas only *wrbA* was differentially expressed between the CK and MAC groups (Table 3).

**Table 2.** Classification of differentially expressed genes between the control and inositol groups.

| Pathway | Gene | Gene Function |
|---|---|---|
| Amino sugar and nucleotide sugar metabolism | E3.2.1.14<br>UGDH<br>galK | chitinase<br>UDPglucose 6-dehydrogenase<br>galactokinase |
| Glutathione metabolism | GST<br>HPGDS<br>RRM2 | glutathione S-transferase<br>glutathione transferase<br>ribonucleoside-diphosphate reductase subunit M2 |
| Pentose and glucuronate interconversions | UGDH<br>E3.2.1.67<br>GAAA<br>adh | UDPglucose 6-dehydrogenase<br>galacturan 1,4-alpha-galacturonidase<br>D-galacturonate reductase<br>alcohol dehydrogenase (NADP+) |
| Purine metabolism | RRM2<br>rdgB<br>add<br>ppnN<br>uaZ | ribonucleoside-diphosphate reductase subunit M2<br>XTP/dITP diphosphohydrolase<br>adenosine deaminase<br>purine-5′-nucleotide nucleosidase<br>urate oxidase |
| Ubiquinone and other terpenoid-quinone biosynthesis | COQ2<br>ARO8 | 4-hydroxybenzoate polyprenyltransferase<br>aromatic amino acid aminotransferase I |
| Starch and sucrose metabolism | E3.2.1.4<br>AMY<br>E3.2.1.58<br>E2.4.1.34 | endoglucanase<br>alpha-amylase<br>glucan 1,3-beta-glucosidase<br>1,3-beta-glucan synthase |
| Valine, leucine, and isoleucine degradation | IVD<br>ALDH | isovaleryl-CoA dehydrogenase<br>aldehyde dehydrogenase (NAD$^+$) |

Statistical analysis of significant differences showed that the expression of two genes associated with ubiquinone and other terpene-benzoquinone biosynthesis genes, *COQ2* and *ARO8*, were downregulated in IO compared with CK group, and *wrbA* expression was also downregulated in the MAC group compared to that in the CK group (Table 4). The chitosanase encoded by *E3.2.1.14* was upregulated in IO and downregulated in MAC compared to that in the CK groups. The functions of *CAT*, catalase, and glucan 1,3-beta-glucosidase, encoded by *E3.2.1.58*, were significantly downregulated in CK and MAC groups. The functions of UDP glucose 6-dehydrogenase, isovaleryl-CoA dehydrogenase, and α-amylase were significantly up-regulated in the *UGDH*, *IVD*, and *AMY* genes of IO compared with CK.

**Table 3.** Classification of differentially expressed genes between the control and maleic acid groups.

| Pathway | Gene | Gene Function |
|---|---|---|
| Steroid biosynthesis | ERG2<br>CYP51 | C-8 sterol isomerase<br>sterol 14alpha-demethylase |
| Tryptophan metabolism | CAT<br>amiE | catalase<br>amidase |
| Ubiquinone and other terpenoid-quinone biosynthesis | wrbA | NAD(P)H dehydrogenase (quinone) |
| Amino sugar and nucleotide sugar metabolism | E3.2.1.14<br>abfA | chitinase<br>alpha-L-arabinofuranosidase |
| Glyoxylate and dicarboxylate metabolism | CAT | catalase |
| Starch and sucrose metabolism | E3.2.1.58<br>malZ | glucan 1,3-beta-glucosidase<br>alpha-glucosidase |
| Fructose and mannose metabolism | MAN | mannan endo-1,4-beta-mannosidase |

**Table 4.** Transcriptome sequence analysis of differentially expressed genes of interest in the inositol and maleic acid groups compared to the control group.

| Gene Name | KO | Gene Function | log2FoldChange (IO/CK) | log2FoldChange (MAC/CK) | *p*-Value (IO/CK) | *p*-Value (MAC/CK) |
|---|---|---|---|---|---|---|
| wrbA | K03809 | NAD(P)H dehydrogenase (quinone) | - | −1.45603 | - | 0.01894 |
| E3.2.1.14 | K01183 | chitinase | 2.644593 | −1.66418 | $1.56 \times 10^{-51}$ | $7.81 \times 10^{-8}$ |
| CAT | K03781 | catalase | - | −1.2896 | - | - |
| E3.2.1.58 | K01210 | glucan 1,3-beta-glucosidase | | −1.32575 | - | - |
| UGDH | K00012 | UDPglucose 6-dehydrogenase | 1.537218 | - | $4.08 \times 10^{-19}$ | - |
| COQ2 | K06125 | 4-hydroxybenzoate polyprenyltransferase | −1.91367 | - | $6.78 \times 10^{-14}$ | - |
| ARO8 | K00838 | aromatic amino acid aminotransferase I | −1.28393 | - | $7.71 \times 10^{-5}$ | - |
| IVD | K00253 | isovaleryl-CoA dehydrogenase | 1.009587 | - | $1.92 \times 10^{-8}$ | - |
| AMY | K01176 | alpha-amylase | 1.765041 | - | $3.55 \times 10^{-25}$ | - |

### 3.8. Validation of RNA-Seq Data Using qPCR

The mRNA expression levels of genes involved in antrodin C biosynthesis after the addition of IO and MAC were also investigated (Figure 7). The expression levels of two genes (*COQ2* and *ARO8*) encoding 4-hydroxybenzoic acid polypentyltransferase and aromatic amino acid aminotransferase I, were significantly reduced in the inositol group, whereas the expression levels of *UGDH*, *IVD*, and *E3.2.1.14* were expressed at significantly higher levels in the inositol group compared with the CK group (Figure 7A). Notably, the expression levels of *wrbA*, *E3.2.1.14*, *CAT*, and *E3.2.1.58* were significantly lower in the MAC group than the CK group (Figure 7B). Thus, the qPCR results confirmed the RNA-seq results.

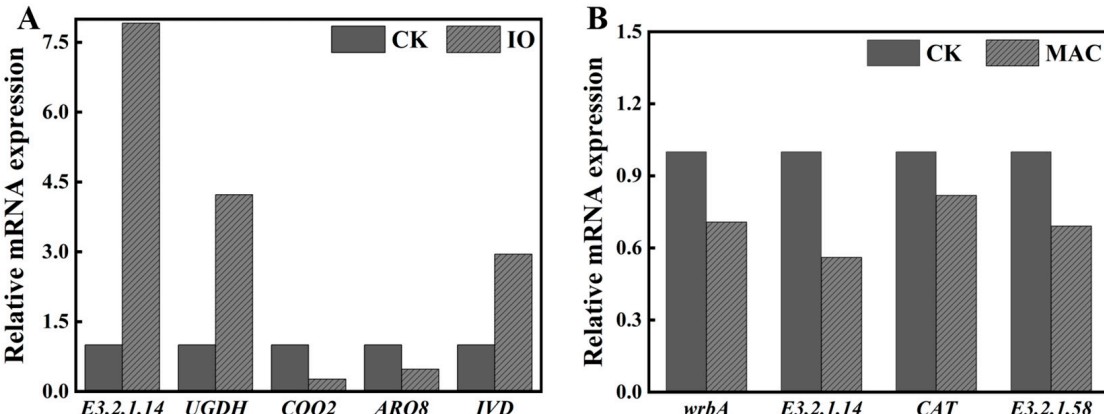

**Figure 7.** Validation of RNA sequencing data using quantitative polymerase chain reaction. Analysis of the mRNA expression levels of genes in the inositol (IO) (**A**) and maleic acid (MAC) (**B**) groups that may be related to the biosynthesis of antrodin C.

## 4. Discussion

Antrodin C is recognised as one of the biologically active metabolites in *T. camphorates* [19], and submerged fermentation culture with the addition of exogenous additives or precursors is considered to be the most efficient method for the industrial production of antrodin C. Hu et al. [20] induced the production of large quantities of the target by adding the potential precursors coenzyme Q0 and p-hydroxybenzoic acid to a submerged liquid fermentation of *T. camphorates*. In our previous study, non-carbocyclic compounds were found to be more susceptible to feedback inhibition than carbocyclic compounds as fermentation precursors [16]. Therefore, we chose concentration gradients of MAC and inositol to explore the changes in antrodin C production. We found that the addition of 0.1 g/L MAC highly significantly inhibited the production of antrodin C compared to the control culture conditions. In contrast, the production of antrodin C was maximised at an inositol concentration of 0.15 g/L. Therefore, these two concentrations were selected for subsequent experiments.

Understanding how *T. camphoratus* differentially expresses metabolites in response to different precursors or exogenous additives during submerged culture is a complex problem. To study the biosynthetic mechanism of antrodin C production by *T. camphoratus*, transcriptome sequencing was conducted. After quality filtering, 4–4.94 million clean reads were obtained. Of these reads, 94.40–96.17% were uniquely mapped to the malls reference genome of *T. camphoratus*, and 3.83–5.60% were mapped to multiple loci. In the inositol and MAC groups, more than 8000 and 1000 differential genes, respectively, were categorised in GO categories, suggesting that a wide diversity of transcripts was represented in the dataset. Therefore, these sequence resources are sufficient to provide adequate support for future studies performing transcriptional analyses of *T. camphoratus*.

KEGG pathway analysis of the transcriptome data provided insights into the secondary metabolite biosynthetic pathways. KEGG analysis showed that pathways involved in the carbohydrate metabolism were significantly enriched in the inositol and MAC groups. Amino and nucleotide sugar metabolites play multiple roles in higher plants, particularly in cell wall synthesis and damage repair [21]. Chitinases, often known as pathogenesis-related proteins in plants, play a dual role. They inhibit pathogenic fungal growth by cell wall digestion and release pathogen-borne factors that induce further defence reactions in host plants [22–24]. In the present study, *E3.2.1.14*, which encodes chitinase, was significantly upregulated in the inositol group and downregulated in the MAC group, probably because it initiates chitin expression when inositol is used as an additive to degrade chitin and uses it as a carbon or nitrogen source. In contrast, the downregulation of *E3.2.1.14* may lead to restricted regulation of secondary metabolites in *T. camphoratus*, which may affect antrodin C synthesis. UDP-glucose 6-dehydrogenase (*UGDH*) is an important enzyme

involved in the biosynthetic pathway and is a key step in glucose metabolism. This enzyme catalyses the conversion of UDP-glucose into UDP-glucuronic acid, an intermediate in the glycolate pathway [25]. UDP-glucuronic acid is necessary for the synthesis of a wide range of biologically active substances, including polysaccharides, cell wall components, antibiotics, and secondary metabolites. High expression levels of *UGDH* indicate the increased synthesis of this enzyme by *T. camphoratus* to meet the demand for specific metabolites, thereby maintaining the synthesis of antrodin C in a highly active state. Additionally, *E3.2.1.58* encodes glucan 1,3-beta-glucosidase, an enzyme involved in the degradation of glucan, which catalyses glucan endocytosis to produce oligosaccharides and glucans. The downregulation of this gene expression in the MAC group may have occurred because the addition of cis-butenedioic acid triggers a feedback inhibitory mechanism that inhibits the activity of the antrodin C synthesis pathway. This feedback inhibition may be achieved by downregulating the expression of glucan 1,3-beta-glucosidase. Compared with the CK group, many genes involved in glucose metabolism were downregulated in the MAC group. This suggests that the addition of MAC may inhibit the glucose metabolic pathway and cause cells to switch to other metabolic pathways, thus reducing antrodin C production.

Ubiquinone, a reduced form of ubiquinol, displays antioxidant activity by scavenging reactive oxygen species produced by plant cells in response to stress and pathogen attack, thereby preventing DNA damage and lipid peroxidation [26,27]. P-hydroxybenzoate polyprenyl transferase, which is also referred to as the "CoQ2 enzyme," mediates the second step in the final reaction sequence of coenzyme Q biosynthesis, namely the condensation of the polyisoprenoid side chain with 4-p-hydroxybenzoate [28]. *ARO8* encodes an aromatic amino acid transaminase involved in a key step in the synthesis of ubiquinone and other terpene quinones. It catalyses the transamination reaction between aromatic amino acids (e.g., tyrosine and phenylalanine, a precursor of tyrosine) and $\alpha$-ketoglutarate. This reaction results in the transfer of the amino group of an aromatic amino acid to $\alpha$-ketoglutarate, forming the corresponding keto acid [29]. The *wrbA* gene is related to NAD(P)H dehydrogenase (quinone), which converts NADH to $NAD^+$, releasing two electrons. *wrbA* may respond to environmental stress when multiple electron transfer chains are compromised or when the environment is highly oxidised, which may favour antrodin C synthesis. Antrodin C is a triquinane-type sesquiterpene with potent inhibitory activity against the hepatitis C virus [9,30]. The significant downregulation of inositol- and MAC-related ubiquinone genes may be attributed to the fact that the addition of inositol and cis-butenedioic acid leads to the activation of the antrodin C biosynthesis pathway. Cells can regulate the synthesis of ubiquinone by repressing the expression of ubiquinone-synthesis-related genes to maintain metabolic homeostasis.

In summary, we determined that the addition of different additives induced differential gene expression in *T. camphoratus*. Carbohydrate and sugar metabolism pathways were significantly upregulated when inositol was used as an exogenous additive, whereas the addition of MAC may inhibit the glucose metabolism pathway and direct cells to other metabolic pathways, leading to a decrease in antrodin C production. The metabolic pathways and regulatory mechanisms associated with antrodin C synthesis can be revealed by analysing the synthesis of antrodin C upon the addition of MAC and inositol. This is of great significance for revealing the details of the synthetic pathway and the regulatory mechanism of antrodin C. The findings of this study also provide new methods and ideas for further development and utilisation of the biological activity of antrodin C.

**Supplementary Materials:** The following supporting information can be downloaded at: https://www.mdpi.com/article/10.3390/fermentation10010028/s1, Figure S1: Hierarchical cluster analysis of gene expression in three samples; Table S1: Primers used for quantitative polymerase chain reaction; Table S2: The fragments per kilobase of transcripts per million fragments (FPKM) values of *Taiwanofungus camphoratus* genes; Table S3: Control vs. inositol group Gene Ontology enrichment; Table S4: Control vs. maleic acid group Gene Ontology enrichment; Table S5: Control vs. inositol group Kyoto Encyclopedia of Genes and Genomes enrichment; Table S6: Control vs. maleic acid group Kyoto Encyclopedia of Genes and Genomes enrichment.

**Author Contributions:** All authors conceived and designed the analysis. W.-H.W. and J.-S.Z. are the corresponding authors. W.J. and S.-P.G. are equivalent contributing authors. X.-H.L. provided the updated database used. W.J. and S.-P.G. conducted the analysis and wrote the first draft of the paper. The revised version of our manuscript was contributed by J.-S.Z. and W.-H.W. All authors have read and agreed to the published version of the manuscript.

**Funding:** This work was supported by the Agriculture Research System of Shanghai, China (Grant No. 202409), and the SAAS Program for Excellent Research Team (Grant No. G2022003).

**Institutional Review Board Statement:** Not applicable.

**Informed Consent Statement:** Not applicable.

**Data Availability Statement:** All data are contained within the article and the Supplementary Materials.

**Acknowledgments:** The authors thank Jian-Wei Ju, Shou-Bing Zhang, and Bao-An Ding of Shanghai Tongyan Industrial Co., Ltd., Shanghai, for valuable assistance with this study.

**Conflicts of Interest:** The authors declare no conflict of interest.

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
