# Peer review of "Transcriptional Analysis of Antrodin C Synthesis in Taiwanofungus camphoratus (Syn. Antrodia camphorate, Antrodia cinnamomea) to Understand Its Biosynthetic Mechanism"

_fermentation, doi:10.3390/fermentation10010028_

Round 1

Reviewer 1 Report

Comments and Suggestions for Authors

I would like to offer my comments and suggestions regarding the manuscript. The experiment has been meticulously planned and presented in the manuscript titled "RNA-Seq Transcriptional Analysis of Antrodin C Synthesis in Taiwanofungus camphoratus for understanding its Biosynthetic Mechanism" in a commendable manner.

Here are rephrased suggestions for improvement:

  1. The authors should consider revising the article for English grammar and spelling errors, as several phrases are challenging to comprehend. The novelty of the article remains unclear to me.
  2. Regarding the title, it would be beneficial to specify the scientific name of the lettuce studied for clarity.
  3. It is advised to pay close attention to punctuation, commas, spacing, and capitalization throughout the entire manuscript.
  4. The introduction section requires a more detailed explanation of the problem, supported by references, to elucidate what sets this study apart.
  5. The conclusions need to be lucid and explicitly highlight the novelty presented in the article.

Thank you for considering these suggestions. I believe these enhancements will significantly contribute to the manuscript's overall quality.

Best regards,

Comments on the Quality of English Language

I would like to offer my comments and suggestions regarding the manuscript. The experiment has been meticulously planned and presented in the manuscript titled "RNA-Seq Transcriptional Analysis of Antrodin C Synthesis in Taiwanofungus camphoratus for understanding its Biosynthetic Mechanism" in a commendable manner.

Here are rephrased suggestions for improvement:

  1. The authors should consider revising the article for English grammar and spelling errors, as several phrases are challenging to comprehend. The novelty of the article remains unclear to me.
  2. Regarding the title, it would be beneficial to specify the scientific name of the lettuce studied for clarity.
  3. It is advised to pay close attention to punctuation, commas, spacing, and capitalization throughout the entire manuscript.
  4. The introduction section requires a more detailed explanation of the problem, supported by references, to elucidate what sets this study apart.
  5. The conclusions need to be lucid and explicitly highlight the novelty presented in the article.

Thank you for considering these suggestions. I believe these enhancements will significantly contribute to the manuscript's overall quality.

Best regards,

Reviewer 2 Report

Comments and Suggestions for Authors

The manuscript “RNA-Seq Transcriptional Analysis of Antrodin C Synthesis in Taiwanofungus camphoratus for understanding its Biosynthetic Mechanism” by Jia et al sent for publication to Fermentation deals with important topic such as Antrodin C  synthesis in an important medicinal fungus T. camphoratus. The manuscript will be of interest to the scientific community working on the topic. Below is the evaluation report.

Minor remarks:

-        all gene names should be in italic (ex. L26)

-        please check the reference list and correct it according journal’s instruction (ex. L162, L480, L482…)

Introduction:

The introduction is somehow well written but can be improved. L54-L57 is more like materials and methods.

Materials and methods:

M&M are well written. I just have a question about WGCNA. Did the authors use all the RNA-seq data (all the transcript identified) for the co-expression analysis?

Results:

This part is well written but can be improved. I think the authors can add the overall identified genes after RNA-seq. Information about T. camphoratus genome size and sequencing should be added. In Figure 3, I don’t see SD.

Discussion:

This section is well written and supported by results. But, I have a question, L356: the authors wrote “…more than 8,000nand 1,000 different genes…” Are the authors sure about that numbers?

Overall, the manuscript deserved to be published after addressing the above-mentioned comments.

Comments on the Quality of English Language
